# Design of Quercetin-Loaded Natural Oil-Based Nanostructured Lipid Carriers for the Treatment of Bacterial Skin Infections

**DOI:** 10.3390/molecules27248818

**Published:** 2022-12-12

**Authors:** Dragana P. C. de Barros, Rafaela Santos, Patricia Reed, Luís P. Fonseca, Abel Oliva

**Affiliations:** 1Instituto de Tecnologia Química e Biológica António Xavier, Universidade Nova de Lisboa, 2780-157 Oeiras, Portugal; 2Department of Bioengineering, Institute for Bioengineering and Biosciences, Instituto Superior Técnico, Universidade de Lisboa, Avenida Rovisco Pais, 1049-001 Lisboa, Portugal; 3IBET, Instituto de Biologia Experimental e Biológica, Apartado 12, 2781-901 Oeiras, Portugal

**Keywords:** NLCs, natural plant oils, quercetin, stability, biocompatibility, antimicrobial activity

## Abstract

The biological activity of natural plant-oil-based nanostructured lipid carriers (NPO-NLCs) can be enhanced by the encapsulation of bioactive compounds, and they in turn can improve topical delivery of the drugs. Quercetin (QR), a vital plant flavonoid, expresses antibacterial properties, and we recently showed that empty NPO-NLCs also have antimicrobial activity. The main objective of this study was to evaluate the synergetic effect of loading natural plant-oil-based nanostructured lipid carriers with quercetin (QR-NPO-NLCs) as a topical delivery system for the treatment of bacterial skin infections. Five nanostructured lipid carrier systems containing different oils (sunflower, olive, corn, coconut, and castor) were engineered. The particles’ stability, structural properties, bioavailability, and antimicrobial activity were studied. NLCs with an average size of <200 nm and Z-potential of −40 mV were developed. Stable QR-NPO-NLCs were obtained with high encapsulation efficiency (>99%). The encapsulation of QR decreased cytotoxicity and increased the antioxidant effect of nanocarriers. An increase in antibacterial activity of the systems containing QR was demonstrated against *Staphylococcus aureus*. QR-NPO-NLCs could transport QR to an intranuclear location within HaCaT cells, indicating that QR-NPO-NLCs are promising candidates for controlled topical drug delivery.

## 1. Introduction

The skin provides a critical barrier against bacterial infections [1,2]. Topical antimicrobial treatments are commonly used to treat skin infections. They provide antimicrobial activity specifically against the causative agent, have a high and continued drug concentration at the infected site, have fewer systemic side effects, and reduce the development of antibiotic resistance [3,4,5]. The most common bacterial skin pathogen is *Staphylococcus aureus*, and its high levels of antimicrobial resistance (AMR) makes it one of the biggest global public health threats today. The development of new antimicrobial approaches such as treatment of AMR strains with bioactive compounds is therefore urgently needed [6,7].

Polyphenols (PPhs) are naturally occurring phytochemicals from a group of secondary metabolites found in all vascular plants, with versatile therapeutic potential. PPhs can interact with proteins, enzymes, and membrane receptors depending on their structure and physicochemical features [8,9,10]. Topical and transdermal delivery of PPhs can provide clinical benefits for certain skin diseases and conditions (e.g., inflammation, wounds, skin cancer, and premature aging) [11,12]. Quercetin (QR; 3,5,7,30,40-pentahydroxyflavone; Figure 1) is considered one of the most expressive PPhs with broad therapeutic potential, and is found in numerous plants. Various studies have reported that the presence of multiple hydroxyl groups, placed at the C 3-, 3′-, 4′-, 5-, and 7-positions in quercetin may be responsible for its biological activity [13]. Promising therapeutic applications of QR on the skin are attributed to its beneficial effects, such as antioxidant, anti-inflammatory, anticancer, antiaging, and antibacterial properties [14,15,16,17]. Additionally, QR has also achieved GRAS (Generally Recognized As Safe) status by the United States Food and Drug Administration [18].

QR’s antibacterial activity has been related to its solubility and interaction with the bacterial cell membrane, changing the cell permeability, due to the presence of numerous hydroxyl groups. [19]. Wang et al. reported a stronger bacteriostatic effect of QR on Gram-positive bacteria than on Gram-negative bacteria [20]. In addition, the synergic effect of QR in combination with other bioactive compounds and antibiotics to inhibit the growth of bacteria was reported [21,22]. Unfortunately, although it has excellent therapeutic potential, quercetin has low bioavailability and poor percutaneous absorption, making it unsuitable for dermal delivery [13].

QR encapsulation into nanoparticles has been proposed to increase its bioavailability, by increasing solubility and physicochemical stability, enhancing skin permeability, and providing a controlled release of the drug [23,24]. For example, Feng Li et al. [25] demonstrated that QR-loaded chitosan-based particles strongly suppressed the growth of *Eschericia coli, S. aureus* and *Bacillus subtilis* compared to native drugs. Different lipid-based delivery systems have been reported as the most effective carrier systems for QR encapsulation [26,27,28]. QR solubility studies showed increased solubility in oily phases, probably distributed at the oil/tensioactive interface [29,30].

Nanostructured lipid carriers (NLCs) are composed of biocompatible and biodegradable lipids. They have gained significant attention as safe and biodegradable systems for dermal drug delivery due to their stability, biocompatibility, and ability to improve drug bioavailability [31]. Imran et al. [32] reported that NLC gels loaded with quercetin and resveratrol increased the delivery of these compounds in the skin layers. NLCs have the potential for topical application of drug moieties due to their good tolerance on the skin and lipid composition, having biological properties such as natural lipids. Pivetta et al. [33] produced quercetin-loaded NLCs to increase skin permeation and obtain a safe and effective topical formulation. These nanoparticles can be produced using natural lipids with biological properties, resulting in a carrier with biological activities [31,34]. Pinto et al. developed stable, non-irritant, natural plant-oil-based NLCs (NPO-NLCs) for encapsulating α-tocopherol and retinoid derivates with improved permeation profiles and with 67% drug loading [35,36,37]. The beneficial impact of natural plant oils (NPOs) on human health is well known, as these ingredients are rich in nutrients, vitamins, minerals, and polyphenols [38,39,40]. Most of these compounds (e.g., free fatty acids and polyphenols) already have antimicrobial activity [11,41]. Our previous work showed the antimicrobial activity of various non-loaded NPO-NLC formulations on *Staphylococcus aureus*, which varied depending on the type of oils used in carrier formulations [41]. All systems demonstrated some level of bacterial toxicity; however, the SF-NLC and OV-NLC systems show the highest antibacterial effect with the smallest cytotoxicity for human dermal cells.

Further evaluation is needed to understand how QR encapsulation affects biocompatibility and antibacterial efficiency. The drug loading capacity of NLCs depends on the solubility of the respective drug in the lipid matrix and is one of the important factors for developing an NLC formulation. The hydrophobic character of QR with a polar group influences its solubility in lipids. The nanostructured lipid carriers (NLCs) based on NPOs and myristic acid (MA) used as a solid lipid (SL) were tested for QR encapsulation (QR-NPO-NLCs). Table 1 presents five oils with different fatty acid composition used as a liquid lipid (LL) in the NLC formulations. Sunflower oil (SF), corn oil (CO), and olive oil (OV) have very similar ratios between saturated and unsaturated fatty acids. Castor oil (CS) is mostly composed of hydroxylated ricinoleic acid (>90%), and coconut oil (CC) is composed of 95% saturated fatty acids.

In the first part of this study, particle size, charge, and physical stability were studied, and quercetin encapsulation efficiency, drug loading, antioxidant activity, and in vitro quercetin release were determined for the NPO-NLC formulations. The second part of this work investigated biocompatibility on keratinocyte cell lines and QR permeability on reconstructed human epidermis (RHE) to examine the interaction between quercetin and its formulations with skin tissue. The antibacterial activity of optimized QR-NPO-NLCs was tested against *S. aureus*.

## 2. Results and Discussion

### 2.1. Physicochemical Characteristics of QR-NPO-NLCs

#### 2.1.1. Physicochemical Properties and Loading Efficiency of QR-NPO-NLC Formulations

The effects of QR loading on the particle size and physical stability of the five NPO-NLCs (NPO: sunflower, olive, corn, castor, and coconut oil) using Span 80 (2%, *w*/*v*) as a surfactant and myristic acid (C14:0) as a solid lipid were evaluated. An excipient selection study by the authors in previous work [41] for the empty NPO-NLCs with SF, CO, OV, CS, and CC showed that the systems with the SL:LL = 40:60 lipid ratios had the best characteristics regarding the stability of the lipid carriers. Therefore, before the biological studies, optimizing QR encapsulation of the systems developed in this work regarding physicochemical stability was necessary. Based on previous results, the SF-NLC system was chosen as a case study for the effect of QR concentration.

The mean particle size, PDI, and zeta potential of QR-SF-NLC formulations with amounts of quercetin ranging from 0.12% (*w*/*v*) to 5% (*w*/*v*) are shown in Figure 1. The particle size was statistically significant and ranged between 160.5 ± 5.9 for SF-NLC without QR and 505.9 ± 31.8 nm for QR-SF-NLCs with 5% (*w*/*v*) QR. A considerable increase in the particle size (>200 nm) was also followed by a PDI increase found for the QR concentration above 1% (*w*/*v*) (Figure 1). A distinct particle size and PDI increase indicate strong system destabilization, probably resulting in aggregation phenomena. A similar behavior was observed with glyceryl dibehenate (Compritol 888^®^, Paramus, NJ, USA) based lipid particle systems by Bose et al. [30] by increasing 10% quercetin loading.

The Z-potential values for all QR concentrations ranged from −51.9 ± 4.7 for the SF–NLC with 0.75% QR to −33.7 ± 1.2 for the 5% QR concentration are shown in Figure 1. The incorporation of QR decreases particle charge. As all systems were >|30 mV|, the repulsion force between particles stabilizes the QR-SF-NLC system. In addition to the high value of Z-potential, a relatively high PDI increase (~60% for the 0.12 QR-SF-NLC) was observed by QR encapsulation with values between 0.289 ± 0.05 for the SF-NLC without QR to 0.658 ± 0.06 for the SF-NLC with 5% of QR. Liu et al. [44] described an increase in PDI from 0.457 to 0.888 for QR-loaded glyceryl monostearate/media chain triglyceride-based NLCs from 0.3% to 0.4% (*w*/*v*), respectively. In addition, by incorporating QR in NLCs, authors found a ~40% increase in PDI. PDI is one of the important factors in the physical stability of nanocarriers. It has been reported that PDI above 0.4 signifies that the system has broad particle size distribution (Appendix A). One possible reason for this could be the high homogenization temperatures, causing aggregation and increasing the particle size distribution [45].

Nevertheless, the physical stability of colloidal dispersions is also predicted by the electrostatic mobility of dispersed nanoparticles (Z-potential). The electrostatic repulsion among the particles that prevented the coalescence of particles, thereby ensuring physical stability of the optimized NLC, was shown by the high negative value of ZP for all tested QR concentrations showing sharp uniform peaks (Appendix A). The PDI of the QR-SF-NLC stays stable for QR concentration ranges between 0.25% (*v*/*w*) and 0.75% (*w*/*v*) with values below 0.5. Regarding size, PDI and Z-potential values of QR concentrations of 0.25%, 0.5%, and 0.75% were tested for all five NPO-NLCs (Figure 2).

QR-NPO-NLCs with 0.5% (*w*/*v*) QR present an average particle size < 200 nm. Nanoparticles between 20 and 200 nm could collect on skin imperfections (pores and follicles), so the drugs from NLCs might more closely interact with the skin, providing better drug penetration through the stratum corneum or the skin [29]. The PDI values stayed below 0.5 for all systems prepared with a QR below 0.5% (*w*/*v*) and Z-potential above −40 mV.

An exception was found for the 0.5% (*w*/*v*) QR-CS-NLC, where the average particle size was higher than the other two observed QR concentrations. This may be because of the castor oil’s high viscosity due to the presence of ricinoleic acid (Table 1), which interferes with experimental reproducibility. Gonzalez-Mira et al. [46] obtained a flurbiprofen (FB)-loaded NLC formulation composed of 0.05 (wt%) FB, 1.6 (wt%) Tween 80, and 50:50 ratios of stearic acid to castor oil, with an average diameter of 288 nm, PI of 0.245, and ZP of −29 mV.

In addition to the different QR solubility in observed NPOs excipients (Appendix A; Supplementary material), encapsulation efficiency (EE) and drug loading for the systems with 0.5% QR did not show any difference. High EE was observed from 99.90% ± 0.01 (QR-CC-NLC) to 99.99% ± 0.03 (QR-CO-NLC). Drug loading capacity (DL) was 16.67 ± 0.1.

All the lipid nanoparticle formulations prepared with each oil demonstrated a high ability to incorporate QR in each of the five tested oils, showing high EE values above 99%. The obtained results for DL demonstrate that the lipid matrices of QR-NPO-NLCs prepared with each natural oil can effectively load quercetin.

#### 2.1.2. Solid-State Characterization of QR-NPO-NLC Systems Using DSC

DSC analysis was performed to evaluate the interactions between the lipids and the loaded active compound, analyzing the crystallinity state of empty NPO-NLCs and QR-NPO-NLCs (Figure 3, Table 2). Figure 3 shows the DSC thermograms of the MA, QR, and QR-NPO-NLCs.

The curves of the QR-NPO-NLCs show two distinct endothermic peaks. The first, between 41.4 °C (QR-OV-NLC) and 50.3 °C (QR-SF-NLC), corresponds to the Tm of MA. QR demonstrated a sharp endothermic peak at 326.5 °C, which shifted distinctly to 315.4, 322.2, 321.7, 302.0, and 312.5 °C in the presence of SF, OV, CO, CC, and CS, respectively. The nature of liquid lipids considerably impacted the depression of Tm of QR. The melting point depression compared with the pure solid lipid, MA (56.3 °C), and QR (326 °C) was observed for all systems (Figure 3).

The liquid lipids’ chemical nature influenced the solid lipids’ melting behavior in the NLC formulations [41,47]. Comparing the DSC results for QR-NPO-NLCs prepared with different oils with those obtained for free NPO-NLCs (Table 2), a decrease in the melting temperatures (Tm, °C) along with a reduction in the energy of lipid modification with the exception for systems with castor oil were observed. The castor oil system showed an increase in the Tm peak of MA that can be attributed to the physical structure of the castor oil. The composition of castor oil has >90% ricinoleic acid, an unsaturated hydroxy fatty acid (C18:1), which could interact with the polar group of QR and change its solubility. In the QR-CC-NLC formulation, the melting peak of QR is almost not detectable. It was shown that QR has 25- and 27-times higher solubility in castor oil (1.6 mg mL^−1^) than in coconut oil (63 µg mL^−1^) and myristic acid (µg mL^−1^), respectively (Appendix A). Solubility tests for SF, OV, and CO oils showed very low solubility. Moderate changes in Tm (°C), ΔH (J/g), and CI (%) indicate the low impact on the crystalline structure of loaded in comparison with non-loaded lipid particles. Non-loaded CS-NLCs already showed a high amorphous structure, so QR incorporation and high solubility within CS oil may not change its crystallinity. This could be due to the presence of QR in an amorphous state or a molecularly dispersed state within the lipid matrix [30].

The resulting enthalpies and crystallinity indexes of QR-NPO-NLCs were 52.3%, 32.1%, and 21.6% lower than empty NPO-NLCs for CC-, CO-, and OV-based systems, respectively. These lower percentages of crystallinity and the evident reduction in the energy for lipid modification may indicate an amorphous phase as a physical state of NLCs. In addition, the reduction in peak areas (enthalpy) (Figure 3) also showed a decrease in lipid crystallinity, which could be attributed to the amorphous structure.

Moreover, the reduction in enthalpy, melting temperature, and crystallinity degree from QR-NPO-NLC formulations suggests that quercetin was mainly captured into the lipid core of nanocarriers, affecting the crystallization process of solid lipids.

These results indicate that incorporating QR resulted in more disturbance inside the lipid crystal lattice than with empty NPO-NLCs (Table 2). Depending on the solubility of the encapsulated active compound, crystallization of the melt can result in a monolithic solid solution or a solid dispersion containing the active compound in a homogenous distribution or the formation of clusters [47,48].

The low crystalline indexes obtained indicated that most likely the NLCs solidified after cooling but did not recrystallize, instead remaining in an amorphous state.

The melting temperatures of the NLCs produced are higher than 40 °C (Table 2), thus the NLCs are solid at the average human body temperature, a condition for their topical administration [49].

#### 2.1.3. FTIR Spectra of QR-NPO-NLCs

FTIR analysis (Bruker IFS66/S, Bruker Optics, Ettlingen, Germany), is one of the important tools for the quick and efficient identification of encapsulated chemical molecules. To identify the interaction between the QR and the NPO-NLC matrix, the FTIR spectra of pure QR, MA (solid lipid), NPO-NLCs, and QR-NPO-NLCs were analyzed, as shown in Figure 4.

The principal characteristic peaks of myristic acid [50] of 2848–2916 cm^−1^ due to symmetrical and asymmetrical stretching, respectively, of the -CH_2_ functional group in fatty acids, C=O stretching vibration at 1701 cm^−1^, and -OH vibrations within the range of 680–940 cm^−1^ due to the aliphatic chain of MA were present in both systems, QR-NPO-NLCs and NPO-NLCs (Figure 4a,b).

FTIR spectra of QR-loaded NLC nanoparticles show peaks of quercetin associated [32] with –OH group stretching (3400–3500cm^−1^), –C=C aromatic bending and stretching (1100–1600 cm^−1^), and –OH phenolic bending (1200–1400 cm^−1^) (Figure 4a). These peaks are absent in NPO-NLC nanoparticles (Figure 4b).

The shifting and modification of the characteristic peaks of the individual components in the spectrum of QR-NPO-NLCs (Figure 4a) demonstrate the indirect confirmation of the incorporation of the QR within the newly formed nanostructure.

#### 2.1.4. Storage Stability of QR-NLCs

The particle size and zeta potential are crucial elements for estimating the stability of the colloidal delivery systems during storage. Therefore, the QR-NPO-NLC formulations with 0.5% quercetin loading were stability tested for six months at room temperature. The particle size, charge, and PDI were measured to assess the physical stability of the systems at specific time intervals during the 6-month period (Figure 5).

As shown in Figure 5, compared with the initial particle size, a low variation in the particle size (<7%) was observed for the five different formulations kept at room temperature for six months.

In addition to the relatively high initial PDI (~0.5), a decrease in PDI values was observed for all systems. The highest PDI decreases of 11.5%, 21.9%, and 28.4% were observed with the NPO-NLC systems with CO, OV, and CC oils, respectively. This may be due to the stabilization effect, where initially formed aggregates probably precipitate, and flocculation or aggregation is decreased during the storage. The high storage stability could be related to the high negative electrical charge of zeta potential values (above −40 mV, Figure 5), which indicates an appropriate electrostatic repulsion between the particles and predicts good physical stability under the time of storage [51]. All these data showed the high stability of quercetin co-loaded NLCs during the storage period.

### 2.2. In Vitro Drug Release Studies

Optimized NLC formulations with 0.5% QR were subjected to in vitro release experiments using an equilibrium dialysis method. The release of quercetin from NPO-NLCs depended on the NPO used in the formulation. The QR encapsulated in the lipid nanoparticles demonstrated a well-controlled release within the first 8 h and a subsequent slow and sustained release phase for up to 72 h (Figure 6). The QR-SF-NLC and QR-CS-NLC exhibited much higher drug release rates than other systems, with cumulative releases at 72 h of 36.3 ± 4.5% and 27.6 ± 2.3%, respectively. The cumulative release values for the QR-CO-NLC, QR-OV-NLC, and QR-CC-NLC systems were 9.6 ± 2.5%, 5.2 ± 2.1%, and 7.0 ± 2.7%. The pure quercetin solution exhibited a much higher release rate than any of the NLC systems, with approximately 63% of quercetin dissolved after 72 h.

One explanation for this is that due to the semi-solid core of the NLC systems, the structure of the NLC part of quercetin might be dissolved in the oil phase, while the other part might be located in the semi-solid core.

The high amount of quercetin dissolved in the liquid shell with a large surface area may account for the fast release at the initial stage, and the encapsulated quercetin in the semi-solid core of NLCs may lead to the sustained release in the later phase [52]. The slower and sustained release of quercetin observed with QR-NPO-NLC systems may be attributed to the diffusion of the quercetin entrapped within the core of the nanoparticles. Souto and Muller [53] reported that the release profile could reflect the encapsulation pattern of the active compound in the lipid matrix. The slow release of QR potentially minimizes the drug’s negative side effects and protects the chemical stability of the bioactive compound.

### 2.3. Antioxidant Effect of QR-NLCs

A DPPH assay was used to estimate the scavenging activity of free quercetin and quercetin-loaded NPO-NLC nanoparticles. The antioxidant properties of pure vegetable oils are related to their protective function in human skin against oxidative stress. The natural oils possess a significant source of antioxidants such as vitamins, minerals, and polyphenolic compounds [38,39]. It was also reported that QR can protect dermal cells from oxidative stress and free-radical-induced toxicity [54,55]. Figure 7 shows the in vitro antioxidant activity in a pure QR solution in each vegetable oil used to prepare the NLCs (SF, CS, OV, CO, and CC), in non-loaded NPO-NLCs, and in QR-NPO-NLCs. This test is based on the ability of DPPH radicals to scavenge oxygenated free radicals. The activity of free quercetin was 96.5 ± 0.21%. The QR-NPO-NLCs with SF, OV, CO, and CS showed high antioxidant activities with percentages of 64.7% ± 7.4, 53.7% ± 2.2, 59.7% ± 6.4, and 66.5% ± 7.9, respectively. QR-CC-NPO showed a lower antioxidant activity (34.6% ± 2.5). For SF, CO, and CS systems, the antioxidant activity of QR-NPO-NLCs was superior to empty NPO-NLCs but lower than that observed with the pure oils. However, a different tendency was observed for OV and CC systems, where the antioxidant activity of QR-OV-NLC and QR-CC-NLC systems was higher than that of empty NLCs and pure oils.

The difference in antioxidant activity between different QR-NPO-NLC systems can be attributed to the variability and structural differences of antioxidant compounds within the oils. Quercetin has strong antioxidant activity due to the presence of multiple hydroxyl groups within its structure [56], while NPOs used in the NLC formulations contain very few hydroxyl groups. Huang et al. demonstrated the lower antioxidant potential of QR in comparison to linseed oil, which contains a small amount of polyphenols in its structure [57]. In the case of the CS-NLC system, the composition of castor oil (>90% unsaturated hydroxy fatty acids) could lead to interactions with the polar group of QR and decrease the number of QR hydroxyl groups responsible for the antioxidative effect. The total antioxidant activity is probably due to a synergetic interaction among polyphenolic compounds [38,58]. QR improves the antioxidant potential of observed systems in comparison with empty NPO-NLCs.

### 2.4. Biological Assays

#### 2.4.1. Cytotoxicity

The main goal of the cytotoxicity studies was to determine the maximum QR-NPO-NLC concentration which showed no cytotoxic effect in HaCaT cell lines yet retained antibacterial activity. Quercetin is a very unstable compound to be used directly for topical therapy. It has been shown that QR has potentially toxic effects, including its mutagenicity, pro-oxidant activity, and mitochondrial toxicity [59]. Encapsulation protects QR from oxidation and degradation and controls its release and contact with the cells. Its antioxidant activity could be responsible for higher cell viability regarding QR-NPO-NLC versus NPO-NLC formulations (Figure 8).

The in vitro cytotoxic effect of free QR, empty NPO-NLCs, and QR-loaded NLCs (QR 5%, *w*/*v*) was assessed on HaCaT cell lines (keratinocytes). A reduction in cell viability of more than 30% was considered a cytotoxic effect, according to ISO 10993-5:2009 [60]. All formulations showed a concentration-dependent effect, with toxicity increasing proportional to the NLC concentration (Figure 8). The non-toxic concentration of NLC-NPO systems for HaCaT cell lines is expressed as the total lipid ratio for free and loaded formulations and is presented in Table 3.

De Barros et al. [41] previously reported low cytotoxic effects for the SF-NLC, OV-NLC, and CO-NLC formulations on HaCaT cell lines (total lipids below 1.25 mg mL^−1^). Quercetin loading in NPO-NLCs increased cell viability for all systems and concentrations compared to free QR and empty NPO-NLCs. For example, in CC systems the minimal non-toxic particle concentration (MNTPC), expressed as the amount of total lipid (mg mL^−1^), increased three times (from 0.833 to 2.5 mg mL^−1^). However, very few reports of cells surviving doses higher than 1 mg mL^−1^ exist, mainly composed of excipients with known safety profiles with a negative surface charge [61,62]. Even for the systems with no difference in MNTPC, such as systems with SF and CO, a higher cell viability was observed for the same concentration level in the case of QR-NPO-NLCs (Figure 8).

Zhu et al. [55] showed that pre-treatment with quercetin significantly increased HaCaT cell viability 24 h post-irradiation. Pivetta et al. also demonstrated that HaCaT cells exposed to QR-loaded natural NLCs increased cell viability compared to those exposed to free QR [33].

Therefore, the NLC formulation containing QR led to a significant decrease in cell toxicity, indicating the possibility of using higher particle concentrations for increased antibacterial activity.

#### 2.4.2. Cellular Uptake of QR-NLCs by HaCaT Cells

The cell nucleus is the main target site for many therapeutic drugs [63]. In previous work, the authors demonstrated the perinuclear location of eight NPO-NLC systems [41], which indicates they have the capacity to deliver higher drug concentrations to the nucleus. Additionally, this work suggested the potential use of NLC systems to deliver encapsulated drugs to specific intracellular targets without causing damage to the cells. Based on the drug release study (Section 2.2), the QR-SF-NLC system was chosen for cellular uptake analyses on HaCat cells. The uptake of a non-toxic concentration (1:50) of QR-SF-NLCs by HaCat cells after a 3 h exposure was observed by Confocal Scanning Laser Microscopy (CSLM) (Figure 9). For comparison, the SF-NLCs were labeled with DiO and incubated with HaCaT cells under the same conditions (Figure 9c).

CLSM images show that HaCat cells cultured with non-labeled lipid nanoparticles displayed no green fluorescence (Figure 9a). When HaCat cells were incubated with free QR, which has autofluorescence, intracellular uptake of the molecule was seen and concentrates in specific structures within the nucleus were observed (Figure 9b). G. Notas et al. [64] reported that in HepG2 and breast cancer cell line T47D, internalized quercetin was predominantly present in the nucleus, suggesting that quercetin might (partially) concentrate at nucleoli.

Incubation of HaCat cells with DiO-labeled SF-NLCs confirmed the intracellular localization previously observed for this system [41], the perinuclear area of the cell (Figure 9c). The perinuclear uptake of lipid particles was previously observed in keratinocytes [65,66]. The HaCaT cells treated with QR-SF-NLCs showed the intracellular accumulation of QR in the perinuclear space and within the nucleus, indicating the drug delivery function of the designed carrier (Figure 9d). QR-SF-NLC particles were not labeled with DiO, as quercetin shows autofluorescence, enabling its simple detection inside cells. Using CLSM, we confirmed the cellular uptake of NPO-NLCs under conditions that were shown to be non-toxic to human cells.

### 2.5. In Vitro Permeation and Histology Studies on an RHE Model

An in vitro skin permeation study using reconstructed human epidermis (RHE) was performed to investigate the effect of the different oils on the enhancement of skin permeation of quercetin using the Franz diffusion cell over 24 h.

In vitro permeation profiles of QR-NPO-NLCs showed similar profiles to those observed during the drug release studies with higher permeation rates observed for QR-SF-NLC and QR-CS-NLC systems (Figure 10). According to the results obtained in this work, after the first 8 h, approximately 75% (SF-NLC), 74% (CS-NLC), 74% (CO-NLC), and 68% (OV-NLC) of the systems reached the receptor phase (absolute ethanol: distilled water = 35:65% *v*/*v*). A slower permeation profile was observed for the CC-NLC, where after 8 h only 51% of the cumulative drug amount was reached. One explanation for this is that the lipid matrix of the QR-CC-NLC is made mostly from unsaturated fatty acids, which could influence the drug capture inside of the lipid matrix.

The permeability studies of Tan et al. [67] of 0.5% QR-loaded lecithin–chitosan systems on the dorsal skin of mice previously showed a significant amount of drug in the epidermis layer. They reported that nanoparticles altered the skin surface and the stratum corneum, disrupting the conjugation between the corneocyte layers and resulting in increased QR permeation in the skin.

The cultured skin membranes used in this study were histologically examined to evaluate the ability of NPO-NLCs to improve the passage of QR across the skin barrier (Figure 11). Based on the higher permeability profile (Figure 10), the SF-NLC system was chosen as a case of study. The green fluorescent fluorophore DiO was incorporated into the lipid matrix to allow for the localization of the lipid nanoparticles in the permeation assays, and DAPI (blue) was used to label the nuclei of the viable cells. The SF-NLC-DiO system was seen to be located in upper layer of the stratum corneum (SC), where fluorescence is more intense (Figure 11a). Sunflower-oil-based NLCs do not penetrate the SC, but are responsible for occlusion effects on the skin, which increase the skin permeability, as observed by Pinto at al. [37].

A higher fluorescence intensity on the uppermost layer of RHE and increased green fluorescence throughout other layers was observed for QR-SF-NLC in comparison with empty SF-NLC-DiO. This can be explained by the diffusion of QR, which has autofluorescence, throughout the SC, after being released from the nanoparticles.

The results of the quercetin permeation ability through the skin demonstrated the role of the structure of the NLCs in acting as an absorption promoter.

### 2.6. Antimicrobial Effect NLC-NPO on S. aureus

All five QR-NPO-NLCs were tested for their antimicrobial activity against *S. aureus* strain COL. There is extensive literature concerning the antibacterial effects of various natural oils on *S. aureus* [R70]. As previously mentioned, all chosen oils had different but comparable FFA contents (Table 1), and the solid lipid component of all NPO-NLC formulations was a fixed parameter. Quercetin was loaded at a concentration of 0.5% (*w*/*w*). Wang et al. previously demonstrated the bacteriostatic effect of 0.0085 µM QR in DMSO in vitro on *S. aureus* [20]. In our previous work, [41], we observed that the empty SF-NLC and OV-NLC systems showed antimicrobial activity at concentrations that were shown to be non-toxic for HaCaT cells.

QR loading increased the antibacterial effect of all QR-NPO-NLCs compared to empty NPO-NLCs. The results indicate that the increase in antimicrobial activity, expressed by total lipid concentration, was greater for the QR-SF-NLC (decrease from 6.25 to 0.78 mg mL^−1^) and QR-OV-NLCs (from 3.13 to 0.78 mg mL^−1^). These two systems already showed high antibacterial potential [41].

Figure 12 shows *S. aureus* growth in the presence of CC-NLC and QR-CC-NLC. By QR loading, the antibacterial concentration of the NLC decreased from 6.25 to 0.16 mg mL^−1^, expressed by total lipid concentration, which decreases cytotoxicity of the QR-CC-NLC regarding HaCaT cells.

Encapsulation of QR increases the antibacterial activity of NPO-NLCs, indicating its potential for treatment of skin infections.

## 3. Materials and Methods

### 3.1. Materials

Solid lipid (SL): myristic acid (MA), C14:0 (98%) was purchased from Alfa Aesar (Haverhill, MA, USA). Liquid lipid (LL): Sunflower (SF) oil, corn oil (CO), and olive oil (OV) (Gallo, Portugal) were food-grade commercial products. Castor oil (CS) from *Ricinus communis* (F.J. Campos, Portugal) and virgin coconut oil (CC) (Fauser Vitaquell, Hamburg, Germany) were cosmetic-grade products produced by cold pressure. Span 80 (Sorbitan monooleate, HLB 4.7) and quercetin dihydrate (QR), (97%, *w*/*w*) were bought from Alfa Aesar (Haverhill, MA, USA). The aqueous phase of mini-emulsions was prepared using Milli-Q grade water. Cell lines: human immortalized keratinocytes (HaCaT), human epidermal keratinocytes, neonatal (HEK), the cell media reagents, DMEM (Dulbecco’s modified Eagle’s medium), fetal bovine serum (FBS), trypsin 0.25%, Pen Strep (10,000 U/mL penicillin, 10 µg/mL streptomycin), Trypsin-EDTA (0.25%), phenol red, phosphate-buffered saline (PBS) 1X Solution,, pH 7.4, and the reagent MTT (3-(4,5-dimethylthiazol-2-yl)-2,5-diphenyltetrazolium bromide) were purchased from Gibco, ThermoFisher Scientific (Waltham, MA, USA). The fluorophore for NLC staining, ‘DiO’, DiOC18(3) (3,3′-dioctadecyloxacarbocyanine perchlorate) was bought from Marker Gene Technologies, Inc. (Eugene, OR, USA). Cell stain 4′,6-diamidine-2′-phenylindole dihydrochloride (DAPI) was purchased from Bertin (BioReagent, Montigny le Bretonneux, France) and Wheat Germ Agglutinin Conjugates (WGA, Alexa Fluor 594 conjugate) from Invitrogen ThermoFisher Scientific (Waltham, MA, USA). Formalin solution (neutral solution (neutral buffered 10%) was obtained from Bio-Optica (Milano, Italy). Trifluoroacetic acid (TFA) suitable for HPLC, ≥99.0% was pursued from Sigma Aldrich (St. Louis, MO, USA). Acetonitrile, absolute ethanol, and methanol were of analytic grade (Alfa Aeser, Haverhill, MA, USA).

### 3.2. Fabrication of Quercetin-Loaded NLCs

Five natural oils (SF, OV, CO, CC, and CS) were chosen for the liquid lipids in the NLC formulations. NLCs with and without QR were prepared by the mini-emulsion methodology using an ultrasonication step [41], and myristic acid as the solid lipid for the encapsulation of QR (QR-NPO-NLC). The aqueous phase was made of the Span 80 solution in Milli-Q water heated to lipid phase temperature. The lipid and aqueous phases were heated to 80 °C, which was above the melting point of the solid lipid used. Thus, the lipid and aqueous phases were mixed. The pre-mini-emulsion was homogenized by stirring for 2 h at 300 rpm and then fully homogenized with a Sonifier (Branson 450D, Danbury, CT, USA) for 10 min (10 s on/5 s off, 55% amplitude). The resultant nano-emulsion was then cooled to room temperature and stored. Each NLC formulation was prepared and tested in triplicate.

### 3.3. Nanoparticle Physicochemical Characterization

The hydrodynamic diameter of the NLC particles and polydispersity index (PDI) of the formulations were determined by dynamic light scattering at 25 °C and scattering angle of 173° (Zetasizer^®^Nano ZS, Malvern PCS Instruments, Malvern UK) 24 h after preparation. The samples were added to a cuvette without dilution before the measurement. Particle sizes and PDIs are given as the average of three measurements. The equipment’s controlling software performed data processing, and the particle size data were evaluated using the intensity distribution.

Zeta (Z)-potential of NLCs was measured by electrophoretic mobility using the same equipment. The analyses were conducted at 25 °C, and the samples were diluted with Milli-Q water (1:10, *v*/*v*). The reported values are the mean ± standard deviation (SD) of at least three different batches of each NLC formulation.

Statistical analysis of variance for particle size, PDI, and ZP was performed with Microsoft Excel 2013 software (Microsoft, Redmond, WA, USA) by normal distribution using a significance level of α = 0.05.

### 3.4. Lyophilization of NPO-NLCs

Samples of NPO-NLC were freeze-dried under vacuum using a lyophilizer Edwards Micromodul (BOC Ltd., Crawley, UK). A cooling rate of 1 °C/min was used to pre-cool the sample from room temperature to −50 °C, and the sample was maintained at −50 °C for 24 h.

### 3.5. Crystallinity Studies by Differential Scanning Calorimetry (DSC)

Differential scanning calorimetry (DSC) analysis was performed to analyze the crystalline state of QR-SF-NLC with 0.5% QR. The thermograms were recorded using a DSC Q200 F3 (TA Instruments Inc., New Castle, DE, USA). A nitrogen purge provided an inert gas atmosphere within the DSC cell at a 50 mL min^−1^ flow rate. A constant cooling rate of 10 °C/min was applied. Approximately 5–6 mg of dried NLC-NPO sample was hermetically sealed into standard aluminum pans. An empty pan was used as a reference. The samples were equilibrated at 0 °C and then submitted to a heating cycle from 0 to 350 °C with 10 °C/min heating rate. The melting points (Mp, °C) and enthalpies (ΔH, J g^−1^) were evaluated using the TA Universal Analysis 2000 (v4.5.0.5) software ((TA Instruments Inc., New Castle, DE, USA). The determination of the crystallinity index (CI, %) was calculated as follows:$$CI,\;\left(\%\right)=\frac{\Delta H_{NLC-NPO}}{\Delta H_{MA}}\times 100\;$$
where Δ*H_NLC-__NPO_* and Δ*H_MA_* are the enthalpies of fusion of the NLCs and myristic acid, respectively.

### 3.6. Fourier Transform Infrared Spectroscopy (FTIR) Properties

The FTIR spectra of lyophilized quercetin, myristic acid, NLC nanoparticles, and quercetin-loaded NLC formulations were recorded on KBr plates in the scanning range of 600–4000 cm^−1^ using FTIR Bruker IFS66/S (Bruker Optics, Ettlingen, Germany). Each KBr disc was scanned 45 times at a resolution of 2 cm^–1^. The characteristic peaks were recorded and compared with that obtained with individual excipient used.

### 3.7. Encapsulation Efficiency (EE) and Drug Loading (DL)

The encapsulation efficiency (EE) and the drug loading (DL) were measured indirectly by determining the amount of unentrapped QR in the supernatant after centrifugation. Ultracentrifugation of NLCs was performed at 55,000 rpm for 60 min at −4 °C. Then, the supernatant was collected and filtered through a Millipore membrane filter (0.2 µm), diluted with ethanol, and analyzed using the validated HPLC method. EE and DL of quercetin in QR-NPO-NLC were calculated according to the following equations:$$EE=\frac{W_{t}-W_{f}}{W_{t}}\times 100$$
$$DL=\frac{W_{t}-W_{free}}{W_{t}-W_{f}+Wl}\times 100$$
where *W_t_* is the total mass of QR added to the whole system, *W_f_* is the mass of free QR determined in the dispersion medium, and *Wl* is the mass of the lipid phase in the NLC formulation. The reported results are the mean ± SD of at least three different batches of each NLC formulation.

The quercetin content in the supernatant was detected by HPLC Waters Alliance 2695 HPLC Separations Module (Dublin, Ireland) using Waters Symmetry C18 (100 Å, 5 µm, 4.6 mm × 250 mm) column. The mobile phase consisted of acetonitrile and 0.1% TFA (35:65, *v*/*v*). The injection volume was 50 µL, and the flow rate of 1.0 mL min^−1^ was carried out and detected at 370 nm with the column temperature maintained at 30 °C. QR retention time was 5.3. The linearity range of calibration curve was found to be 0.1–500 µg mL^−1^, while the correlation coefficient was 0.995.

### 3.8. In Vitro QR Release

In vitro release studies of QR from NLCs were carried out with dialysis bag method. Dialysis membranes with Molecular Weight Cut Off (MWCO) 12,000–14,000 (Viskase, Lombard, IL, USA) were used for this study. A mixture of absolute ethanol and distilled water in the ratio of 35:65% *v*/*v* was used as the release medium to ensure sufficient sink condition [30]. Each membrane bag containing 5 mL of QR-NPO-NLC suspension was immersed in 200 mL of receptor medium consisting of a mixture of absolute ethanol and bi-distilled water 35:65% *v*/*v*. These were maintained at 37 °C with magnetic stirring at 120 rpm in a closed container to prevent evaporation [37]. The receptor medium was collected in 1 mL aliquots at time intervals (5, 10, 15, and 30 min and 1, 2, 3, 4, 5, 6, 7, 8, 24, 32, and 48 h) and replaced with the same volume of fresh medium. In addition, the release profile of QR control solutions (2.5 mg/mL in propylene glycol) was observed, using the same dialysis process as for NLC suspensions. All samples were measured in triplicate. Samples were stored at 4 °C until analysis. The released QR was quantified by HPLC using the conditions described in Section 3.7. The reported values are the mean ± SD of two different batches of each NLC formulation and control solution.

### 3.9. Antioxidant Activity of QR-NOP-NLCs

The free radical scavenging (antioxidant) capacity of free QR, pure SF, OV, CO, CS, and CC were measured by 1,1-diphenyl-2-picrylhydrazyl (DPPH) assay [37]. An amount of 200 μL of sample solution was added to 100 μL of DPPH solution (0.2 mM) prepared in absolute ethanol. The reaction mixture was incubated at 37 °C for 30 min. The absorbance was measured at 517 nm in a multi-well plate reader (SpectraMax 340PC Microplate Reader, Molecular Devices, LLC., San Jose, CA, USA) against the DPPH control solution. The radical scavenging activity was calculated using the following equation:$$Scavenging\;activity,\;\%=\frac{Ac-As}{Ac}\times 100$$
where *Ac* is the absorbance of control at λ = 517 nm, after a period of 30 min, and as is the absorbance of sample at λ = 517 nm, 30 min after incubation in the presence of antioxidant. All determinations were performed in triplicate and the results are given as the mean ± SD.

### 3.10. Biological Studies

#### 3.10.1. Biocompatibility of QR-NPO-NLCs

The cytotoxicity tests were performed for two dermal cell lines: immortalized human keratinocytes (HaCaT) and human dermal fibroblasts, neonatal (HDFn). HaCaT and HDFn were cultured in 175 cm^2^ flasks using Dulbecco’s modified Eagle’s medium (DMEM) supplemented with 10% fetal bovine serum (FBS) and 0.1% Pen Strep (10,000 U/mL penicillin, 10 µg mL^−1^ streptomycin). Cells were maintained at 37 °C in a 95% air/5% CO_2_ atmosphere and were detached with a trypsin solution (Trypsin/EDTA Solution, Gibco™). The cell lines were harvested at 80% confluence and were seeded in each well of 96-well plates at a density of 2 × 10^4^ cells/well. Cells were grown for 24 h at 37 °C in a 95% air/5% CO_2_ atmosphere to obtain subconfluence. Cells were then washed with a PBS solution and subsequently put in contact with 200 μL of each sample. NLC samples were diluted with DMEM medium in 1/10, 1/20, 1/50, and 1/100 *v*/*v* ratios. For the positive control, cells were kept in contact only with the culture medium. The cytotoxicity of the developed formulations (NLCs) was evaluated, 3 h after exposure, by the (MTT) reduction assay. MTT is a tetrazolium dye that is converted into formazan by metabolically active cells [68]. Briefly, 10 µL of the 12 mM MTT was added to each well with the sample and incubated in a humidified 5% CO_2_-95% air atmosphere for 3 h. After this incubation period, the samples were removed from the wells, washed with PBS to remove unreacted MTT, and the formed formazan crystals dissolved in 100% dimethyl sulfoxide (DMSO) for 10 min. The absorbance was measured at 540 nm in a multi-well plate reader (SpectraMax 340PC Microplate Reader, Molecular Devices, LLC., San Jose, CA, USA). The cell viability (%) was calculated using the following equation:$$Cell\;viability,\;\%=\frac{Absorbance\;of\;treated\;cells}{Absorbance\;of\;negative\;control}\times 100$$
where the negative control was the cells incubated with DMEM medium alone.

#### 3.10.2. Cultivation of Reconstruction of Human Epidermis (RHE)

Primary human epidermal keratinocytes isolated from neonatal foreskin (HEKn; Gibco) were maintained in keratinocyte growth medium (KGM) composed of EpiLife medium (Gibco), supplemented with 0.06 mM calcium and keratinocyte growth factor (HKGS, Gibco), at 37 °C and 5% carbon dioxide (CO_2_) in a humidified incubator.

The development of RHE on cell culture inserts with polycarbonate membranes (Merck, Millipore) was performed following the procedure described by Zoio et al. [69]. The polycarbonate membranes were selected for HKEn attachment without matrix. In brief, the inserts were placed in six-well plates containing 2.5 mL of HEK growth medium with high calcium concentration (1.5 mM) and seeded with 3 × 10^5^ HEKns in 500 mL KGM medium. After 24 h, the models were raised to the air–liquid phase. The medium was replaced by 1.5 mL KGM supplemented with 1.5 mM calcium, 50 mg mL^−1^ l-ascorbic acid 2-phosphate, and 10 ng mL^−1^ keratinocytes growth factor (KGF) (San Diego, CA, USA). This culture medium was replaced every 48 h and after 11 days at the air–liquid interface, until the RHE was morphologically fully differentiated.

#### 3.10.3. Permeation Study Using Franz Diffusion Cells

In vitro permeation studies of QR-NPO-NLC dispersions were performed using engineered static Franz diffusion cells, designed and constructed at the Biomolecular Diagnostic Laboratory at Instituto de Tecnologia Química e Biológica da Universidade Nova de Lisboa (ITQB NOVA, Oeiras, Portugal). The diffusion cells have an effective diffusional area of 1.45 cm^2^ and a chamber capacity of 13.9 mL to properly accommodate the inserts used to support the RHE.

The study was conducted according to the conditions established in the OECD Guideline 428 (2004) [70]. The inserts supporting the formed RHE were mounted on the receptor compartment of the Franz diffusion cells. The diffusion cells were filled with the receptor medium (14 mL) and were left to hydrate for 1 h in a water bath at 37 ± 0.5 °C under constant stirring at 250 rpm. The receptor medium consisted of a solution of absolute ethanol: distilled water (35:65% *v*/*v*) with sink conditions. After the hydration, 500 μL of the NLC dispersion was added into the donor compartment. Aliquots of 1 mL were withdrawn at established time intervals (0, 5, 15, 30, 45, and 60 min and 1, 2, 3, 4, 5, 6, 7, 8, and 24 h) and replaced with the same volume of fresh receptor medium.

The experiments were performed in triplicate. All samples were collected in vials and kept at −20 °C until analyzed by HPLC as described in Section 3.7.

#### 3.10.4. Histological Analyses

Tissue staining and histological analyses were conducted according to our previous study [41,69]. For the histological analysis of the permeation studies, the inserts were removed from the Franz diffusion cells and the surface supporting the RHE was washed with 1 mL PBS, pH 7.4 1X Solution, to remove residual QR-SF-NLCs. The samples were fixed immediately after being taken out of culture in 10% neutral-buffered formalin (Sigma-Aldrich, St. Louis, MO, USA) for a minimum of 24 h at room temperature.

Prior to cryopreservation, the RHEs on the polycarbonate filters were detached from the inserts and immersed in 30% sucrose to cryoprotect the tissues. Cryofixation of the tissues was followed by cryostat sectioning. Skin sections (5 mm thick) were mounted on slides for histological analysis. Some tissue sections were conventionally stained with hematoxylin–eosin staining to allow a standard morphological analysis of the RHE, while others were hydrated and stained with DAPI for fluorescent DNA labeling of cell nuclei. Images were obtained using the Nikon Eclipse TE2000-S fluorescence microscope (Nikon instruments, Melville, NY, USA) and analyzed with the ImageJ Software (National Institutes of Health, Bethesda, MD, USA).

#### 3.10.5. Confocal Laser Scanning Microscopy (CLSM)

For CLSM imaging, cells were prepared on coverslips inside 6-well plates at a density of 1 × 10^5^ cells per well and incubated for 24 h at 37 °C and 5% CO_2_. HaCat cells were then incubated with DiO-labeled empty NPO-NLCs OR non-labeled QR-NPO-NLCs for 3 h at 37 °C, 5% CO_2_. After incubation, cells were washed twice with PBS and fixed with 10% formalin for 10 min at room temperature. After washing twice with PBS, coverslips were stained with 5 µg/mL of WGA-Alexa 594 for 10 min at 37 °C. Cells were washed twice with PBS and nuclei were stained with 300 nM DAPI for 5 min at room temperature, then washed twice more with PBS. Coverslips were then inverted over microscope slides using Vectashield (Vector Laboratories Burlingame, CA, USA) as a mounting medium for fluorescence photobleaching prevention. CLSM images were acquired on a Leica SP5 CLSM (Leica Microsystems, Wetzlar, Germany) and processed using a Leica Application Suite—LAS AFv4.3 software (Leica Microsystems, Wetzlar, Germany).

#### 3.10.6. Antibacterial Studies

Antimicrobial properties of NLCs were assessed against *Staphylococcus aureus* reference strain COL (Methicillin-resistant *S. aureus* isolated from infection, common lab strain). The minimum inhibitory concentration of NLCs was determined using the microdilution method [71]. Briefly, 100 μL of each NPO-NLC was added to Well 1 with 100 μL Tryptic Soy Broth (TSB, Difco) bacterial growth medium, then serially diluted in Well 10 in a 96-well plate. Well 11 served as a control with no NPO-NLC added and Well 12 as a sterility control. Each well (1–11) was then inoculated with *S. aureus* (5 × 10^3^ cells), and the plates were incubated at 37 °C, with constant shaking, for 16 h. OD_600nm_ was measured every 30 min in a plate reader BioTek Synergy Neo2 (BioTek U.S., Winooski, VT, USA). Each experiment was repeated in triplicate.

### 3.11. Statistical Analysis

Results were expressed as mean ± SD. Statistical analysis of variance for in vitro release studies was performed in Microsoft Excel^®^ 2013 software by Normal Distribution using a significance level of α = 0.05, the average of each experiment in the five Franz diffusion cells (μ) from two independent assays, and its standard deviation (σ).

## 4. Conclusions

In this study, five natural oils were selected and successfully applied in new formulations of multifunctional QR-loaded NLCs. The QR concentration used influenced the physicochemical stability of the NLCs by changing the diameter of the NLC formulations (between 160 nm and 185 nm) and the Z-potential (between −46 mV and −61 mV). In addition, the degree of crystallinity moved to a more amorphous structure, indicating a good QR capture.

The antioxidant activity of QR-NPO-NLCs was increased when compared to that of the free NPO-NLCs, reaching higher values for the SF- and CO-NLCs systems.

The cytotoxicity of the QR-NPO-NLC on HaCaT cells decreased compared to that observed for NPO-NLC systems, indicating the benefit of encapsulating QR. Using CLSM, we confirmed the cellular uptake of QR-NLC-NPOs around and inside the nucleus under conditions that were shown to be non-toxic to human cells.

Permeation studies indicated slow, controlled drug diffusion using the RHE model.

The results indicate that QR-NPO-NLC systems have higher antimicrobial activity than free NPO-NLCs, which varies depending on the type of lipid carrier. In addition, QR loading showed a decrease in cytotoxic cell concentration at antibacterial concentrations.

This research contributes to the development of safe, natural oil-based lipid nanocarriers with antimicrobial effects using drug-conjugated systems with natural antibacterial compounds, reducing their secondary effects and physicochemical instability.

For future studies, to understand better the impact of lipids on cellular interactions of NLC-NPOs and the transport mechanism, time- and particle-concentration-dependent cellular uptake will be performed and impacts of other bioactive compounds from NPO structure will be analyzed.

## Data Availability

The data presented in this study are available on request from the corresponding author.

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
