# Peer review of "Design of Quercetin-Loaded Natural Oil-Based Nanostructured Lipid Carriers for the Treatment of Bacterial Skin Infections"

_molecules, 2022, doi:10.3390/molecules27248818_

Round 1

Reviewer 1 Report

This manuscript presents the evaluation of nanostructured lipid carriers for a natural anti-bacterial flavonoid, quercetin, using five different natural plant oils. This attempt aimed to synergistically improve topical delivery and anti-bacterial activity for countering skin infections. Formulation stability, bioavailability and biological activity against gram positive Staphylococcus aureus, were all through investigated through rigorous experimental protocols. Presented formulation depicted high drug encapsulation efficiency, improved in vitro skin permeation, and improved safety profile for quercetin as the latter was assessed on skin keratinocytes.

This work can be considered as valuable addition to the field. Publication is recommended following minor suggestions as they are listed below:

1.      Chemical structure of quercetin, as well as main components of NPO-NLC should be presented.

2.      The authors performed dynamic light scattering (DLS) particle size analysis and measurement, Can the DLS graphs be incorporated in the manuscript or supporting information?

3.      Likewise, if possible and available, incorporate zeta potential graphs obtained from the instrument in the manuscript or supplementary data.

4.      The composition of mentioned market product Compritol® should be stated.

5.      In Table 2, Enthalapy (ΔH) was introduced without much discussion within the context. Authors are advised to thoroughly annotate it and describe its main findings.

6.      Section 2.1.3. FTIR analysis, authors should annotate for the missed OH peak within the quercetin spectrum as possible interaction binding with the NPO-NLC components.

7.      Could the authors explain how the castor oil would affect the quercetin solubility, and in what direction?

8.      Authors could describe more detailed long-term stability analysis, concerning humidity and light exposure status?

9.      Figure 5 is missing PDI legends for three NPO-NLC formulations.

Author Response

This manuscript presents the evaluation of nanostructured lipid carriers for a natural anti-bacterial flavonoid, quercetin, using five different natural plant oils. This attempt aimed to synergistically improve topical delivery and anti-bacterial activity for countering skin infections. Formulation stability, bioavailability and biological activity against gram positive Staphylococcus aureus, were all through investigated through rigorous experimental protocols. Presented formulation depicted high drug encapsulation efficiency, improved in vitro skin permeation, and improved safety profile for quercetin as the latter was assessed on skin keratinocytes.

This work can be considered as valuable addition to the field. Publication is recommended following minor suggestions as they are listed below:

  1. Chemical structure of quercetin, as well as main components of NPO-NLC should be presented.

The authors acknowledge the reviewer's comment and added the chemical structure of quercetin in the Introduction (page 2, lines 45-48).  A description of the main components of NPO-NLC is given on page 3, lines 95-109 and Table 1.

  1. The authors performed dynamic light scattering (DLS) particle size analysis and measurement, Can the DLS graphs be incorporated in the manuscript or supporting information?
  2. Likewise, if possible and available, incorporate zeta potential graphs obtained from the instrument in the manuscript or supplementary data.

The authors acknowledge the comment of the reviewer, and our response is regarding reviewer requests 2. and 3.

The incorporation of DLS and Z-potential graphs are very useful in explaining lower QR-NPO-NLCs emulsion stability regarding NPO-NLCs. PDI was between 0.4 and 0.5 for QR-NPO-NLCs, which indicates moderate emulsion stability.

However, all experiments were perfomed at least in triplicate and the measurement of all samples by Malvern equipment was also done in triplicate. The amount of data, including all graphs for DLS and Z-potential, would be impossible to add to the manuscript. The authors present a summary of DLS graphs and Z-potential graphs, supporting Figure 1 (S1, Supplementary Materials).

Also, the authors added a new text in the manuscript to support additional data – page 4, lines 151-164.

“Nevertheless, the physical stability of colloidal dispersions is also predicted by the electrostatic mobility of dispersed nanoparticles (Z-potential). The electrostatic repulsion among the particles that prevented the coalescence of particles, thereby ensuring physical stability of the optimized NLC, was shown by the high negative value of ZP for all tested QR concentrations showing sharps uniform peaks (S1, Supplementary Materials).”

  1. The composition of mentioned market product Compritol® should be stated.

The authors added on page 4, line 139, the information about the composition of Compritol 888® (glyceryl dibehenate) from Paramus, NJ, USA.

  1. In Table 2, Enthalapy (ΔH) was introduced without much discussion within the context. Authors are advised to thoroughly annotate it and describe its main findings.

The authors acknowledge the comment of the reviewer and add text on page 6, lines 219-228.

“The resulting enthalpies and crystallinity indexes of QR-NPO-NLCs were 52.3 %, 32.1% and 21.6 % lower than empty NPO-NLCs for CC-, CO-, and OV-based systems, respectively. These lower percentages of crystallinity and the evident reduction in the energy for lipid modification may indicate an amorphous phase as a physical state of NLCs. In addition, the reduction of peak areas (enthalpy) (Figure 3) also showed a decrease in lipid crystallinity which could be attributed to the amorphous structure.

Moreover, the reduction of enthalpy, melting temperature, and crystallinity degree from QR-NPO-NLCs formulations suggests that quercetin was mainly captured into the lipid core of nanocarriers affecting the crystallization process of solid lipids. “

  1. Could the authors explain how the castor oil would affect the quercetin solubility, and in what direction?

Authors acknowledge reviewer comment.

The solubility of QR in lipids was determined by lipid solubility studies using by the saturation solubility method. These data were not showed in the original manuscript as authors didn’t see that QR solubility influenced encapsulation efficiency and drug loading. However, we agree that solubility data may be of interest in crystallinity studies behavior so we added text on Page 7, Lines 217-225

“It was shown that the QR has 25 and 27 times higher solubility in castor oil(1.6mg/mL) than in coconut oil (63 µg/mL) and myristic acid (µg/mL), respectively (S2, Table QR solubility in NLCs excipients). Solubility tests for SF, OV and CO oils show very low solubility. Moderate changes of Tm (°C), ΔH (J/g), and CI (%) indicate the low impact on the crystalline structure of loaded in comparison with no loaded lipid particles. No loaded CS-NLS already showed a high amorphous structure, so QR incorporation and high solubility within CS oil may not change its crystallinity. “

  1. Section 2.1.3. FTIR analysis, authors should annotate for the missed OH peak within the quercetin spectrum as possible interaction binding with the NPO-NLC components.

The authors agree with the suggestion of the reviewer and alter the section (Page 7, lines 237-240) as follows:

  “ FTIR spectra of QR-loaded NLC nanoparticles show peaks of quercetin associated [32], with –OH group stretching (3400–3500cm−1 ),  –C=C aromatic bending and stretching (1100–1600cm−1), and –OH phenolic bending (1200–1400cm−1) (Figure 4a). These peaks are absent in NPO-NLCs nanoparticles (Figure 4b).”

  1. Authors could describe more detailed long-term stability analysis, concerning humidity and light exposure status?

The authors acknowledge the comment of the reviewer. However, at this point, stability studies were performed for the NLC nanoemulsion stability. Storage stability of the lyophilized particle forms was not part of this study. The QR-NPO-NLC emulsions were protected from light by aluminium foil, and empty particles were exposed to natural light.

The long-term storage stability regarding the humidity and light exposure status would be separate studies.

  1. Figure 5 is missing PDI legends for three NPO-NLC formulations.

Figure 5 was corrected.

Reviewer 2 Report

The paper prepared and evaluated a series of quercetin-loaded natural oils based nanostructured lipid carriers. However, the following issues should be carefully addressed.

1.     In Section 2.3., “The antioxidant potential of QR-NPO-NLCs was superior to empty NPO-NLCs but lower than for the pure SF, CO and CS.”Why? In addition, the antioxidant potential of QR-NPO-NLCs was lower than QR. Why? Is the amount of QR and lipid the same in each sample? 

2.     In Section 2.4.1., “QR loading in NPO-NLCs increases cell viability for all systems and concentrations compared to free QR and empty NPO-NLCs. ” Why?

3.     Section 2.4.2. indicated that QR-NPO-NLCs can be uptaken by HaCaT cells. Did this character influence the skin permeation and antimicrobial action? Generally, if NLCs were captured by HaCaT cells, the skin permeation would be impaired.

4.     Figure 12 is unclear. 

5.     In vitro test on antibacterial activity is not enough. The animal experiment is suggested. Only in this way, we can judge whether the designed formulation is effective or not.

Author Response

Comments and Suggestions for Authors

The paper prepared and evaluated a series of quercetin-loaded natural oils based nanostructured lipid carriers. However, the following issues should be carefully addressed.

Extensive editing of English language and style of the manuscript was done by a native English-speaking colleague and co-author of this paper Dr. Patricia Reed (https://novaresearch.unl.pt/en/persons/patricia-reed; [email protected]).

In Section 2.3., “The antioxidant potential of QR-NPO-NLCs was superior to empty NPO-NLCs but lower than for the pure SF, CO and CS. ”Why? In addition, the antioxidant potential of QR-NPO-NLCs was lower than QR. Why? Is the amount of QR and lipid the same in each sample? 

The authors acknowledge the reviewer's comment and agree this section needs to be more precise in explaining the antioxidative influence of encapsulation of QR in NPO.

Quercetin has strong antioxidant activity due to the presence of multiple hydroxyl groups in the structure, while NPO used in NLCs formulations contains very little of this structure.  In the case of CS systems, the composition of castor oil ( >90 % unsaturated hydroxy fatty acids) could lead to interactions with the polar group of QR and decrease the number of QR hydroxyl groups responsible for the antioxidative effect.

The authors added a new text and four more references throughout Section 2.3.

  1. In Section 2.4.1., “QR loading in NPO-NLCs increases cell viability for all systems and concentrations compared to free QR and empty NPO-NLCs. ” Why?

The authors acknowledge the reviewer's comment and for more clarity added a new text and one reference in this section.

“The main goal of cytotoxicity studies was to determine no cytotoxic QR-NPO-NLCs concentration regarding HaCaT cell lines which could be used for antibacterial activity. QR is a very unstable compound to be used directly for topical therapy. It has been shown that quercetin has potentially toxic effects, including its mutagenicity, prooxidant activity, mitochondrial toxicity, and inhibition of key enzymes involved in hormone metabolism [Chen et al.,2014]. Encapsulation protects QR from oxidation and degradation and controls QR concentration release and contact with the cells. Its antioxidant activity could be responsible for higher cell viability regarding QR-NPO-NLC.”

Authors already presented in the manuscript the potential of QR to increase HaCaT viability, reported by other authors

  “ Zhu et al. [650] showed that pretreatment with quercetin significantly increased HaCaT cell viability 24 h post‑irradiation. Pivetta et al. also demonstrated that HaCaT exposed to QR-loaded natural NLCs increases cell viability compared to free QR [33].”

  1. Section 2.4.2. indicated that QR-NPO-NLCs can be uptaken by HaCaT cells. Did this character influence the skin permeation and antimicrobial action?

The authors acknowledge the comment of the reviewer.

Through cellular uptake studies, our goal was to show the capability of NPO-NLCs to be effective carriers of bioactive antimicrobial compounds for microbial skin infection. The cellular nuclei are the main target site for therapeutic action. The outmost skin layer, the epidermis, made of keratinocytes, is the first protective barrier against microbial skin infections. For that reason, we carried out cellular uptake studies on these dermal cell lines. The importance of QR delivery to the nuclei of keratinocytes, shown here, showed the potential of QR-NPO-NLCs to treat bacterial invasion on skin superficies. By QR delivery to the nucleus of the cell, QR-NPO-NLCs showed increased antimicrobial potential in comparison with no loaded NPO-NLCs.

The authors agree with the reviewer that QR-NPO-NLCs cellular uptake on antimicrobial impact was not underlined during discussion, so a new text was added:

“QR delivery to nuclei using QR-NPO-NLCs indicates increased antimicrobial potential in comparison with none loaded NPO-NLCs.”

Generally, if NLCs were captured by HaCaT cells, the skin permeation would be impaired.

NLCs permeation profile (Section 2.5) shows the possibility of the carriers to reach the deepest epidermis layers, made from stratified keratinocytes, which would be beneficial to prevent the spreading of skin infection in the dermis. The nanoparticles altered skin surface and stratum corneum and increase QR permeation profile in the skin.

  1. Figure 12 is unclear. 

Figure 12 was corrected.

  1. In vitro test on antibacterial activity is not enough. The animal experiment is suggested. Only in this way, we can judge whether the designed formulation is effective or not.

The authors acknowledge the comment of the reviewer. In this work, the authors aimed to study how antibacterial bioactive compounds change the antibacterial potential of NPO-NLCs. In previous work, authors showed the antibacterial effect of empty NPO-NLCs.

In vitro studies on dermal cell lines (keratinocytes) were previously done. Therefore, the first step was to compare the biocompatibility and antibacterial activity of QR-NPO-NLCs with once-obtained results with free NPO-NLCs on the same dermal cell lines).

As this work indicates an increase in antibacterial effect in QR-NPO-NLCs compared to empty NPO-NLCs, the next step will be in vitro studies of QR-NPO-NLCs bioavailability and antibacterial activity on a 3D full-thickness skin models.

Results obtained from in vitro studies will serve to choose the most promising system(s) to process with animal experiments.